# A Deep Learning-Based Automatic Data Acquisition System for Medical Monitors

Yizhi Zou
*Department of Automation Engineering,*
*University of Electronic Science and Technology of China*
Chengdu, China

Han Cao
*Department of Automation Engineering,*
*University of Electronic Science and Technology of China*
Chengdu, China

Xu Cheng
*Department of Anesthesiology*
*West China Hospital of Sichuan University*
Chengdu, China

Lu Yang
*Department of Automation Engineering,*
*University of Electronic Science and Technology of China*
Chengdu, China
yanglu@uestc.edu.cn

*Abstract*—In the cardiac operating room, several operators are essential to assist the surgeon, including the physician managing and monitoring the artificial heart-lung machine. The custodian must interpret the patient's vital signs from equipment data and make decisions, such as blood transfusion. However, the equipment lacks automated data acquisition and recording capabilities, posing significant challenges for documenting surgical information. This paper introduces a system for screen segmentation and text recognition based on visual methods. This system allows the operating equipment doctor to wear a head-mounted camera to capture real-time video similar to the doctor's perspective, and pulls the video stream of the camera through RTSP (Real-Time Streaming Protocol) on the PC side. We proceed by processing the video stream captured by the camera, leveraging the YOLO (You Only Look Once) algorithm and OCR (Optical Character Recognition) technology as our primary tools to do screen segmentation and text recognition. These technologies enable us to extract the information displayed on the medical equipment screens. By initially employing YOLO for detecting and segmenting the screen of interest, the process is approximately 127% faster than direct OCR processing of the entire video frame. Additionally, the accuracy rate of OCR recognition for clear pictures can also reach more than 95% by our method.

*Index Terms*—real-time system, visual detection, text recognition, data collection

## I. INTRODUCTION

In hospital operating room, it is imperative to document changes in a patient's vital signs and analyze their current physical condition to facilitate appropriate adjustments. Due to the non-unified and proprietary nature of the medical display equipment's transmission interfaces, direct data transmission is not feasible. Consequently, this hinders the data recording and system entry processes. We need to find a suitable way to get the equipment data.

Recently, most monitoring tasks for artificial heart-lung machines rely on traditional manual methods. The main problems include: 1) The method of manual processing of detection data is inefficient and costly. During heart surgery, the results are currently recorded manually with paper and pencil and entered into the hospital system. This model requires repetitive

work and high labor intensity, leading to inefficient data entry. At present, the variety of hospital display equipment lacks uniform standards. There are problems with poor interaction, different data interface protocols and data formats. The low degree of data standardization brings great difficulties to data collection. Data collection costs are high. 2) The authenticity of the test data cannot be guaranteed. At present, the detection data of various medical data display equipment depend on manual recording and input. The integrity and correctness of manually recorded data largely depend on the professional level of field personnel, and the long-term repetitive work can lead to operator distraction, affecting data authenticity.

As technology advances, we are increasingly relying on various screens to obtain information. However, this information is usually only displayed in real-time and cannot be automatically archived, requiring manual monitoring. Although some equipment have data interfaces, the standards and protocols are not unified, making data collection complex and time-consuming. Therefore, collecting screen images through a camera and then transcribing the images into text information is a preferred solution. The system designed in this paper allows the data equipment custodian to carry a portable mobile camera, which can be head-mounted or neck-mounted, to capture real-time video stream from the mobile camera and transmit it to the PC. When the medical signs of patients are recorded in the video, the relevant medical data display equipment is detected and tracked, segment and identified. Finally, the captured data are formatted and transmitted to the hospital's data recording system.

Currently, there is a scarcity of similar engineering work in screen detection and recognition. From the perspective of surgery, most similar work tends to focus on auxiliary processing for surgeons or making informed decisions to facilitate intelligent emergency medical services[1-3]. From the perspective of monitor screen extraction, Yan proposed a vision-based method to extract monitor screens[4]. liu et al. used OCR technology to obtain data from substation equipment screens[5]. The system proposed in this paper is designed from

the perspective of equipment operators to facilitate auxiliary surgery, automatic data collection, and comprehensive record-keeping.

Challenge 1: The dynamic camera will bring more difficulties than the static camera, such as the improper angle of the picture caused by the operator wearing the camera, the picture often rotates, the blur caused by the picture shake, the operating room interference noise and other factors, which brings huge challenges to the stability and reliability of our system.

Challenge 2: Deep learning faces challenges in real-time video processing, necessitating segmentation, tracking, correction, text recognition, and other processing of the video stream transmitted by the camera.

Challenge 3: information transmission challenge, the existing wireless transmission methods include RFID (Radio Frequency Identification), bluetooth, 802.11, and other things, relatively speaking, the short-distance wireless LAN (Local Area Network) transmission rate is faster, more suitable for our system and the hospital's system connection.

Our contributions:

1. We have developed a comprehensive, stable, and reliable system that fully automates system functions and information transmission, reducing the hospital's operational concerns.

2. We have created a custom dataset utilizing YOLOv8, enabling the detection, segmentation, and tracking of specific targets, followed by correction and OCR recognition of identified images. This integration of multiple functionalities into a cohesive system ensures both stability and real-time performance.

3. The construction of wireless network is more convenient, better use of operating room space, and does not affect the normal execution of surgery.

Our system significantly reduces the time equipment managers spend on manually recording data into the hospital system, especially when the operator is temporarily absent. The system avoids processing invalid information through image recognition technology, which automatically collects and analyzes detection data from the display screens of monitors, thereby enhancing the efficiency of data collection. Future iterations of the system could incorporate information processing and decision-making functionalities.

## II. BACKGROUND AND RELATED WORK

In contemporary surgical settings, a lead surgeon requires the support of numerous assistants, nurses managing sterile instruments, and physicians who adjust equipment and monitor patient vital signs. The operating room is a high-pressure environment where staff juggle multiple tasks, often leading to a rushed workflow. Moreover, they must manually input and record the patient's physical signs on the equipment, which can disrupt the surgical process and potentially impact patient outcomes, such as delayed blood transfusions or infusions. To address these challenges, the system presented in this paper is specifically designed to assist in normal operations for reducing the workload of assistants and automating the recording

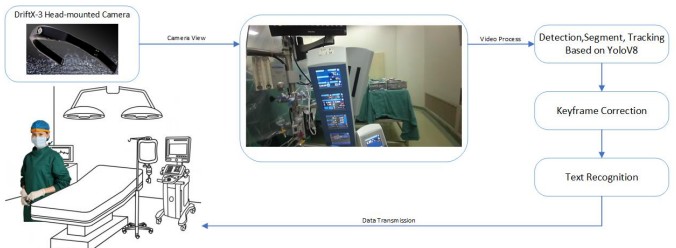

Fig. 1. Overall system framework

and transmission of physical sign data from device screens to the hospital's internal system. The system's framework and its operational process are detailed in Fig.1.

### A. Methods for Screen Detection

Convolutional Neural Networks (CNNs) are a class of deep learning models extensively utilized in image recognition and processing. They function by mimicking the human visual system to identify patterns and features within images. CNNs consist of a multi-layered structure, encompassing convolutional layers, pooling layers, and fully connected layers. The convolutional layers are tasked with extracting image features, while the pooling layers reduce the spatial dimensions of these features. The fully connected layers then perform the final classification. In object detection, CNNs employ algorithms such as R-CNN and its variants (FastR-CNN and FasterR-CNN) to achieve rapid and precise object recognition. These algorithms initiate by generating candidate regions, followed by feature extraction using CNNs, and finally determine the target categories and exact locations through the use of classifiers and regressors. Technological advancements have significantly enhanced the speed and accuracy of CNNs in detection tasks, consolidating their importance in the field of computer vision.

The You Only Look Once (YOLO) series of models encapsulate numerous state-of-the-art (SOTA) technologies into a unified framework. As depicted in Fig.2, the YOLO model works in a very simple way. A single convolutional network concurrently predicts multiple bounding boxes and class probabilities for those boxes. The YOLO series is renowned for its real-time and fast performance in object detection, offering significant advantages in the real-time processing of streaming video.

The YOLO model is extensively employed in various applications such as segmentation, detection, tracking and

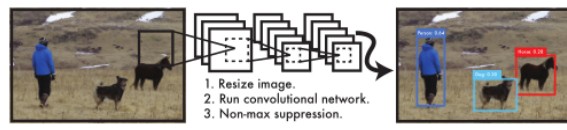

Fig. 2. The YOLO detection system. Processing images with YOLO is simple and straightforward. (1) resizes the input image to 448 × 448, (2) runs a single convolutional network on the image, (3) thresholds the resulting detections by the model's confidence[6]

other aspects, and it has a wide range of applications in clinical medical operating rooms. Including real-time monitoring, disease detection, auxiliary decision, data automatic recording, and postoperative evaluation and research. It can detect key instruments and tissues during surgery in real time, providing precise feedback and tips to help doctors perform operations more accurately and improve the safety and efficiency of surgery. At the same time, YOLO can also be used to automatically generate surgical logs, assist the training of novice doctors, analyze surgical videos to improve surgical techniques and promote the development of clinical medicine. For example, cancer diagnosis based on medical images[7-9]. Accurate and efficient object detection algorithms are essential for assisting healthcare professionals in diagnosing and treating various medical conditions [10-12]. The real time performance of YOLO makes it particularly appealing for time-sensitive medical procedures and clinical decision-making[13].

### B. Methods for Screen Recognition

There are more and more applications of deep learning technology in the medical field, and the representative one is the application of OCR (Optical Character Recognition) technology based on deep learning. OCR technology has two main applications in the medical field, one is used for electronic medical records, and the other is used for drug identification. With the in-depth development of information technology and the growth of the economy, the medical field has an increasingly high demand for electronic medical records. On the one hand, medical institutions will produce a lot of paper documents, including patient medical records, laboratory reports, charging documents and medical insurance policies. Patients need to convert paper documents into electronic versions when handling matters such as insurance claims and hospital transfers. On the other hand, hospitals need to automate the identification and database entry of medical prescriptions or medical records issued to patients[14].

Deep learning has driven significant advances in OCR technology, mainly using technologies such as Convolutional Neural Networks (CNNs), Recurrent Neural Networks (RNN), Connectionist Temporal Classification (CTC), attention mechanisms, and transformer. These methods improve the accuracy and efficiency of character and text recognition, and are widely used in practical applications such as document digitization, license plate recognition, handwriting recognition and ticket processing. In recent years, deep learning and OCR technologies[15- 17] have developed rapidly, providing a new idea for automatic data acquisition. Therefore, this paper proposes an OCR-based automatic collecting method for testing experiment data, which uses image recognition technology to automatically acquire and parse the detection data from the display screen of the testing instrument, which greatly improves the efficiency of automatic collection of detection data[5].

## III. PROPOSED SYSTEM CONSTRUCTION

This section introduces the overall architecture of the proposed system. The main goal is to build a complete system, which is designed to assist the work of the operating room, automatically recognize the text of some special devices and transmit the required data to the hospital storage system. Our system algorithm framework is shown in the Fig.4. We implement a real-time dual-thread system: one thread processes the video stream, removing cached frames and updating the display with the latest frame, which is then passed to the image processing module. The another thread is responsible for detecting, segmenting, correcting, and recognizing text in the latest frame, with results fed back to the video display and data transmitted to the hospital system. While ensuring user-friendly visualization, it also ensures real-time performance as much as possible.

Camera video stream acquisition. The method we adopt is that the camera acts as the role of the stream server, the PC retrieves the camera's IP address via the local area network and accesses the video stream through the RTSP. The PC side sets the parameters of the video stream such as resolution, bit rate, frame rate.

### A. Screen Detection Based on Yolov8

Computer vision tasks such as image classification, detection and segmentation have become the focus of research. In these tasks, YOLO (You Only Look Once) series algorithms have received a lot of attention for their efficiency and accuracy. YOLOv8 performs well in image classification, detection and segmentation, making it a powerful tool for solving these problems in one stop.

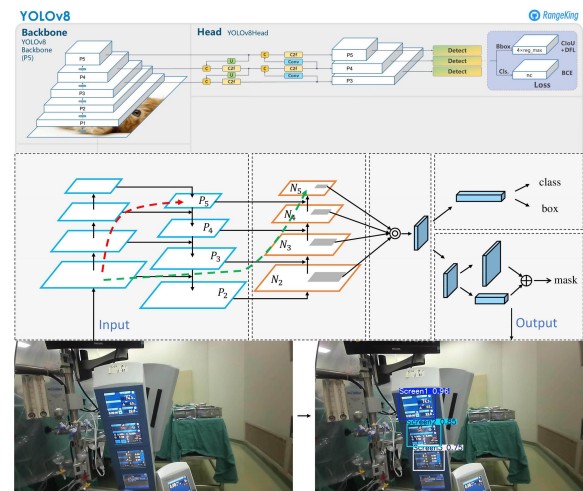

Fig. 3. Network structure of YOLOv8

As Fig.3 shows, YOLOv8 uses an approach called unified framework that can handle image classification, detection, and segmentation simultaneously. The framework uses a backbone network structure, which can extract the feature information in the image. Then, these features are used for different tasks through different head network structures. In terms of image

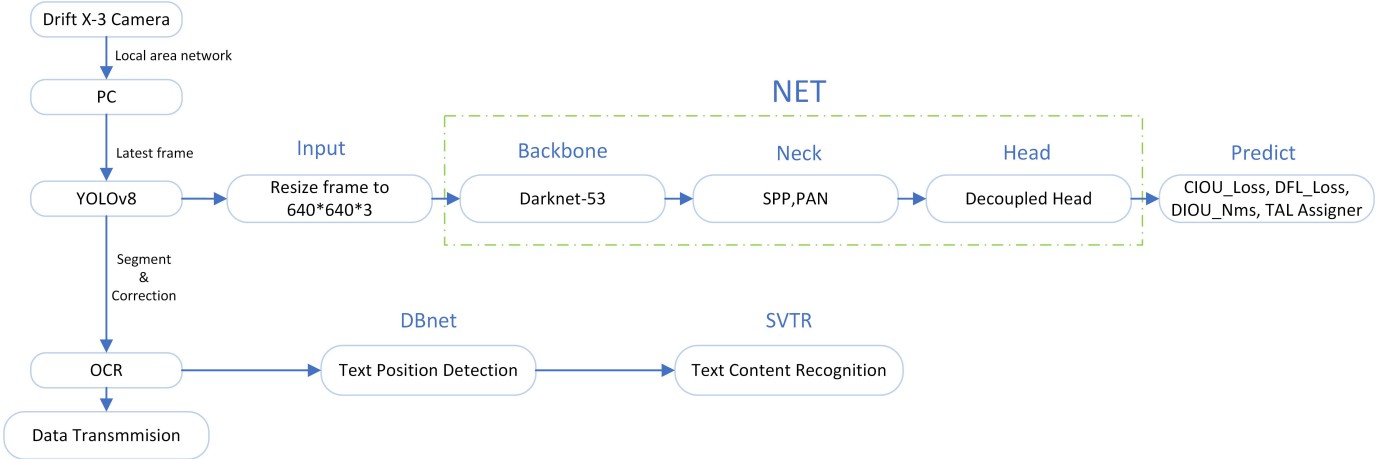

Fig. 4. Algorithm framework of our system

classification, YOLOv8 uses a pre-trained model, which can achieve high accuracy on ImageNet dataset. Owing to the use of the latest technology, YOLOv8 has high accuracy and robustness in image classification. Regarding target detection, YOLOv8 applies multi-scale feature fusion technology, which can simultaneously detect targets of different sizes. In addition, YOLOv8 also introduces a new loss function and anchor frame mechanism to improve the accuracy and speed of target detection. Concerning instance segmentation, YOLOv8 adopts a method similar to MaskR-CNN, which can accurately segment each object in the image. Due to the unified framework, YOLOv8 also has high efficiency and accuracy in segmentation.

The loss function of YOLOv8 includes categorical loss and regression loss, using BCE (Binary Cross Entropy) as the categorical loss. In object detection, YOLOv8 usually needs to predict the presence or absence of the object, so using BCE as a loss function can effectively measure the difference between the predicted result and the real label.

$$
\begin{aligned}
L_{BCE} &= \frac{1}{N}\sum_i L_i \\
&= -\frac{1}{N}\sum_i y_i \cdot \log\left(p_i\right) \\
&\quad + \frac{1}{N}\sum_i \left(1 - y_i\right)\cdot \log\left(1 - p_i\right)
\end{aligned}
\tag{1}
$$

where $p_i$ represents the output prediction result of the model, and $y_i$ represents the real label value. The BCE function calculates a scalar value as a loss value by plugging the predicted result and the true label value into the formula, which is used to measure the error between the predicted result and the true label.

The regression loss CIOU is an improved version of the original IOU (Intersection over Union) loss function. It takes into account the complete intersection between target boxes and introduces correction factors to more accurately measure the similarity between target boxes. The calculation method of the CIOU loss function is more complex than that of traditional IOU, but it also allows the model to better understand the exact position and shape of the target box during training.

$$
L_{CIOU} = 1 - IOU + \frac{d^2}{c^2} + \alpha v
\tag{2}
$$

where $d$ is the distance between the center point of the prediction box and the real box, and $d$ is the diagonal distance of the smallest circumscribed rectangle. Also, $\alpha = \frac{v}{(1-IoU)+v}$ and $v$ is the correction factor and is used to further adjust the loss function, taking into account the shape and orientation of the target box. The specific calculation is as follows:$v = \frac{4\cdot\left(\arctan\frac{w_G}{h_G}-\arctan\frac{w_P}{h_P}\right)^2}{\pi^2}$, $(w_G, h_G)$ and $(w_P, h_P)$ is the width and height of the target and forecast boxes, respectively.

The regression loss introduces DFL (Distribution Focal Loss) in addition to the CIOU loss function because the regression branches need to be bound to the integral form representation proposed in DFL. DFL takes the form of cross entropy to optimize the probability of the two positions closest to the label, one left and one right, thus allowing the network to focus faster on the distribution of the target position and its neighborhood.

$$
\begin{aligned}
L_{DFL}\left(\mathcal{S}_i, \mathcal{S}_{i+1}\right) = &-\left(y_{i+1} - y\right)\log\left(\mathcal{S}_i\right) \\
&+ \left(y - y_i\right)\log\left(\mathcal{S}_{i+1}\right)
\end{aligned}
\tag{3}
$$

where $\mathcal{S}_i$, $\mathcal{S}_{i+1}$ is the network output predicted value, near predicted value. $y$, $y_i$, $y_{i+1}$ is the actual value, label integral value, near label integral value of the label.

The process of converting labels to DFL form is as follows:
1. Convert the label value from $xywh$ to $ltrb$.
2. Calculate the label ltrb four values, and convert to the DFL required integral form (label value + proximity label value).

The specific conversion process is: $y$ = distance from the center of an edge / current downsampling times.

The three Losses are weighted using a certain proportion of weights to give the total loss. Train loss and detection effect are shown in Fig.5 and Fig.6.

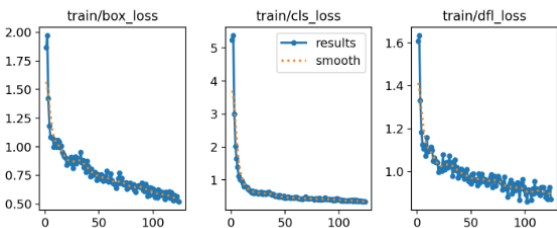

Fig. 5. Train loss

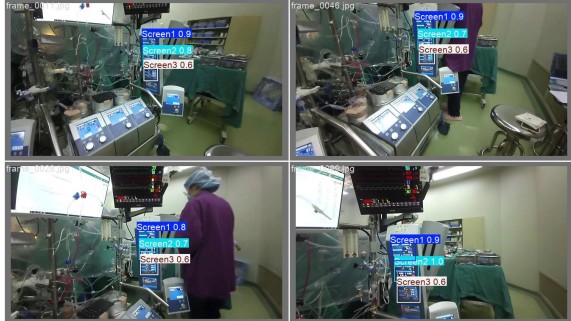

Fig. 6. Detection effect based on YOLOv8

## B. Optical Character Recognition

OCR is composed of two parts, namely text position detection and text content recognition. You need to first detect the contour position of the text, and then further accurately recognize the content.

- Text Position Detection : DBNet

DBNet is a text detection algorithm based on segmentation. It introduces a Differentiable Binarization (DB) module[18], enabling the model to binarize through an adaptive threshold map. This adaptive threshold map can calculate losses, thereby optimizing the model during the training process. DBNet not only enhances text detection results but also simplifies the post-processing procedure, demonstrating significant advantages in both performance and effectiveness.

Compared to the common text detection algorithm based on segmentation, the biggest difference in DBNet lies in its threshold map. Instead of using a fixed value, it predicts the threshold at each position of the image through the network, better separating the text background from the foreground[19]. Fig.7 shows the process of the common text detection algorithm (indicated by the blue arrow) and the DB algorithm (indicated by the red arrow).

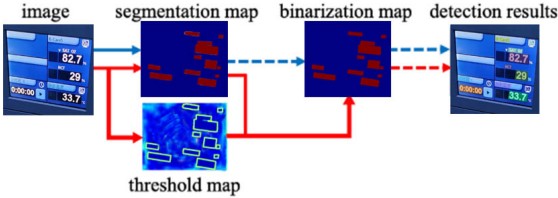

Fig. 7. Text detection algorithm process schematic diagram

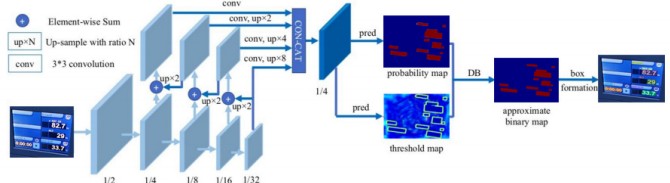

Fig. 8. Overall structure of DB

In traditional image segmentation algorithms, after obtaining the probability map, the standard binarization method will be used for processing, setting the pixel points below the threshold to 0, and the pixel points above the threshold to 1. The formula is as follows:

$$B_{i,j} = \begin{cases} 1 & \text{if } P_{i,j} \geq t \\ 0 \text{ else} \end{cases} \quad (4)$$

where $t$ is the threshold, and $P_{i,j}$ is the probability that the pixel point $(i,j)$ is text.

However, the standard binarization method is nondifferentiable, which prevents the network from being trained end-to-end. To solve this problem, the DB algorithm proposes Differentiable Binarization. Differentiable binarization approximates the step function in standard binarization and replaces it with the following formula:

$$B_{i,j} = \frac{1}{1 + e^{-K(P_{i,j} - T_{i,j})}} \quad (5)$$

where $P_{i,j}$ is the probability that the pixel point $(i,j)$ is text, $T_{i,j}$ is the threshold $T$ for the pixel point $(i,j)$ to be text, and $K$ is the gain factor.

The overall structure of the DB algorithm is shown in Fig. 8. "1/2", "1/4", ... and "1/32" represent the scaling ratio of the feature and the input image.

The input image goes through the network backbone and FPN(Feature Pyramid Networks) to extract features, which are cascaded to get the original image features. The convolution layer generates a text area prediction probability map and a threshold map. DB post-processing obtains the text bounding curve. The content within the curve is cut out for recognition.

- Text Content Recognition : SVTR

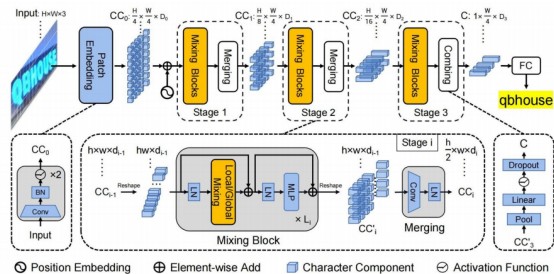

Fig. 9. SVTR model overall structure[20]

This project uses the SVTR model for character recognition. It introduces the transformers structure, effectively mining the

contextual information of text line images to enhance text recognition capabilities[20]. The model inputs an H×W×3 image and obtains character components through Progressive Overlapping Patch Embedding, which are used to represent character strokes. After three stages of downsampling, mixing blocks, merging or combining operations extract features at different scales to generate a representation called C, followed by parallel linear prediction.

As shown in Fig.9, the SVTR model mainly consists of the following parts: Firstly, Progressive Overlapping Patch Embedding is used to divide the image into multiple overlapping patches for feature extraction. Then, the Mixing Block is used for feature mixing, including global mixing to evaluate the dependencies between all character components, and local mixing to assess the correlation between components within a predefined window. Next, Merging combines multiple features into a higher-level feature. Finally, in the Combining and Prediction stage, all features are merged into a feature vector, and a linear classifier is used for prediction to obtain the character sequence.

- Recognition Effect

After the image enters the Text Position Detection model, the model will recognize and output the contours of multiple areas where text may exist. Along with these contours, a confidence inference in the form of a percentage is also output. After deleting the parameters with low confidence, the remaining contours are collected and compared and cut with the original image to obtain a set of images that may contain text.

The image set will be sent to the Text Content Recognition model for text detection after sorting. The detection result will include text IDs and confidence levels. Similarly, information with low confidence is deleted, and the remaining text IDs are compared with the pre-stored dictionary file to restore the text content.

Finally, by combining the text content information and text position information, the recognized text is printed at the corresponding position on the original image for easy viewing by the user, thereby achieving recognition and detection of the screen. The recognition effect is shown in Fig.10.

## IV. EXPERIMENTS AND RESULTS

Hospital extracorporeal circulation machine equipment: The equipment is shown in the Fig.11. The entire Stockert S5 system is used to perform, control and monitor cardiopulmonary bypass during cardiopulmonary perfusion surgery. What we need is to automatically capture information from multiple screens in the equipment and transfer it to the hospital system.

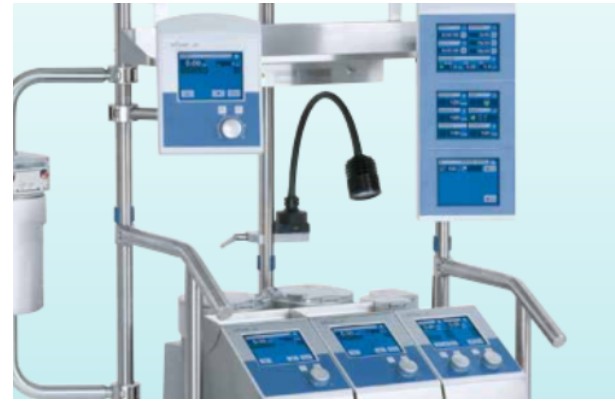

Fig. 11.  Heart-lung machine Stockert S5

Head-mounted camera: DriftX-3, resolution 1080P frame rate 30FPS, video format MP4 (H.264), 120° wide angle, 97 grams, battery life 3 hours.

All experiments is running on lntel(R) Core(TM) i7-9750H CPU @ 2.60GHz.

We collected many pictures and videos in the operating room by using our camera to simulate real wearing conditions, and made a small screen data set. The hospital requires that our system can identify three display screens on the machine equipment, and transfer some of the valid data to the hospital database for storage.

Based on the algorithm model of YOLOv8, we trained the data set made by ourselves and achieved certain results. The training results are shown in the Fig.12, with the loss decreasing continuously and the accuracy improving continuously. After 100 epochs, the model gradually converges, and then the trained model is used to detect and segment the three screens we need, and the results are as follows in Fig.13.

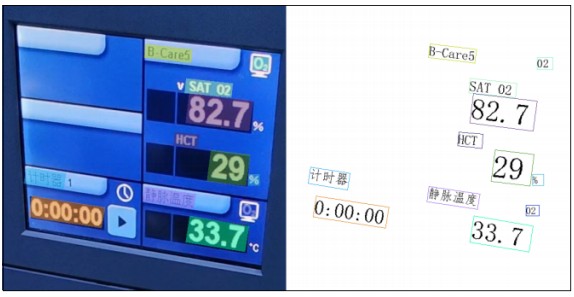

Fig. 10.  OCR recognition effect

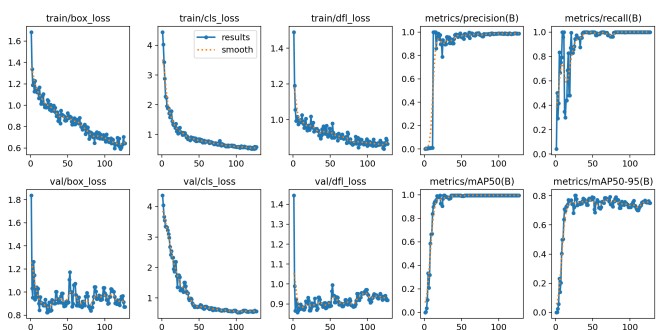

Fig. 12.  Training results

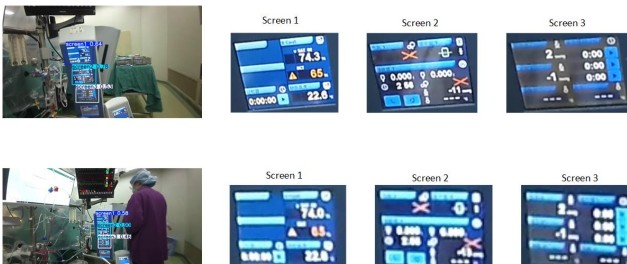

Fig. 13. Detection and segmentation based on YOLOv8

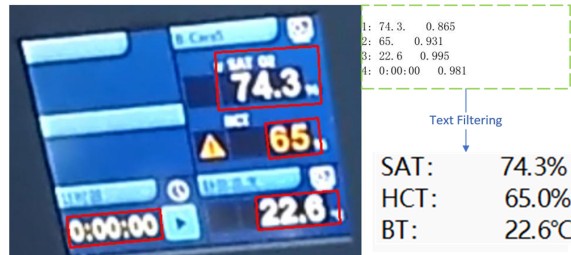

Fig. 15. Text filtering

Table I shows an average class confidence score.

TABLE I
SCREEN DETECTION CONFIDENCE

| Performance Metrics | Screen 1 | Screen 2 | Screen 3 |
|---|---|---|---|
| Average Confidence | 0.88 | 0.83 | 0.6 |

For the segmented images, we perform correction and OCR processing, and compare the impact of correction on OCR processing accuracy, shown in Fig.14. As can be seen from the figure, we can get more text information after OCR processing of the corrected picture.

After OCR processing, we will get a series of text information, but not all of the information is what we need at this time, such as screen 1 we need the information of SAT (Saturation, usually Oxygen Saturation), HCT (Hematocrit), BT (Body Temperature), screen 2 and screen 3 we need other information. Therefore, we also need to do a text screening of information to extract useful information. As Fig.15 shows, we collect the specific data automatically. Additionally, if the camera captures only a portion of the device screen due to reasons such as the screen being out of range, partial detection, unknown obstructions, or reflections, we set the undetected part of the output to null. This ensures that the global operation of the system remains unaffected. If we do OCR processing directly for the picture captured by the camera, it will consume a lot of our text recognition time on invalid information, and

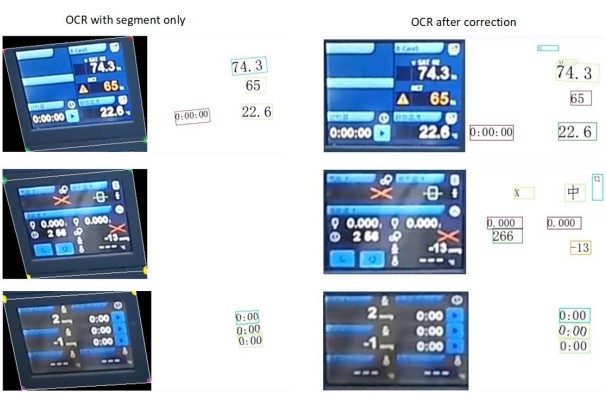

Fig. 14. OCR with correction or not

more noise will also affect the processing of the final result information.

We shot the scene by DriftX-3 camera to imitate the operator's real behavior, and selected 20 pictures for testing, half of which were unclear due to motion blur. In order to obtain real data, we did not abandon these defective pictures. In the first YOLO detection segmentation, two of the image screens were not detected due to extreme blurriness and only part of the screen in the image, and were not included in the statistical analysis of the results. Secondly, in the actual system design, the accuracy and reliability of data are highly required, and the ambiguity caused by motion blur of video stream is inevitable. When we may affect the data result due to some actual external reasons, we need to collect the corresponding data as clearly and accurately as possible. To ensure the reliability of the data, we set that only the data with more than 99% confidence and after screening can be transmitted to the hospital system. On the sheet Table II shows the recognition accuracy and average consumption time of each indicator in screen 1.

TABLE II
PERFORMANCE AT LOW RESOLUTION

| Method | SAT | HCT | BT | Cost(s) |
|---|---|---|---|---|
| OCR with Original Frame | 0.25 | 0.5 | 0.5 | 1.2731 |
| OCR after Segmentation | 0.44 | 0.44 | 0.39 | 0.7 |
| OCR after Segmentation and Correction | 0.56 | 0.5 | 0.5 | 0.56 |

As can be seen from Table II, the processing speed of OCR is faster and the accuracy is also improved to some extent after the detection, segmentation and correction of the acquired images. The main reason for the low accuracy of data recognition in the table is that most of the selected image cameras are not high in definition. The accuracy of OCR recognition can reach more than 95%. In addition, we selected 20 relatively clear and complete data pictures to experiment on our method, and the experiment verified our idea as shown in the table III below:

TABLE III
PERFORMANCE AT HIGH RESOLUTION

| Method | SAT | HCT | BT | Cost(s) |
|---|---|---|---|---|
| Ours | 0.95 | 1 | 0.95 | 0.61 |

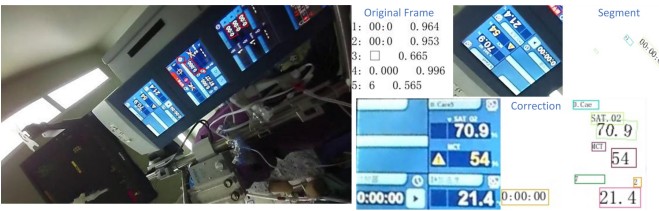

Fig. 16. Correction at a high tilt angle

Compared with the original image without processing, the time required for one detection and segmentation is much less than that required for OCR processing of the whole image. As for whether to correct the image after segmentation depends on the tilt angle of the image, the accuracy and efficiency of OCR recognition are similar in the case that the tilt angle does not change much. However, considering that the operator needs to move around from time to time and the possible movement of the head (tilted head, bowed head), when the tilt angle is high, an image correction in advance can greatly improve the recognition accuracy and efficiency of OCR and it also improves the robustness of the system accordingly. As shown in the Fig.16, we show the effect of whether the tilt angle is corrected or not.

It is obvious from the figure of the experimental results that we cannot recognize the information on the screen before the tilt correction, and the recognition rate has been greatly improved after the correction. All experiments demonstrate the speed and reliability of our system we proposed.

## V. CONCLUSION

In this paper, we designed a system to assist the data collection of some monitors in the operating room, which minimizes the workload of personnel assistants in the limited space of the operating room, and the accuracy and stability of our system is guaranteed. Through some methods of computer vision to help achieve the acquisition of patients' vital signs data, to solve the countless transmission interface or data transmission interface is not unified, not open equipment, and the equipment is in a bad environment is not suitable for manual reading data and other problems. Remote data acquisition through YOLO and OCR technology, leading the transformation and development of the industry to smart metering. Finally, our proposed system combines some segmentation methods and text recognition methods to achieve a more accurate and reliable acquisition of data information in monitors' screens.

### ACKNOWLEDGMENT

We would like to thank the West China Hospital of Sichuan University for providing us with a surgical test site.

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
