# OpenReview forum: "$$A Deep Learning-Based Automatic Data Acquisition System for Medical Monitors$$"
_IEEE.org/ICIST/2024/Conference — IEEE ICIST 2024 Conference Submission_

### Official Review · Reviewer_SRbC · 2024-08-30
**This paper is not the research field of the reviewer.**

**Rating:** 7
**Confidence:** 2

**Review:**

1. Medical equipment displays can vary significantly between manufacturers and models. How does your system account for differences in screen layouts, fonts, and colors? Have you considered training the model on a diverse dataset to improve generalization?

---

### Official Review · Reviewer_p4Qk · 2024-09-01
**This paper designed a system to assist the data collection of some monitors in the operating room, which is attractive for minimizing the workload of personnel assistants in the limited space of the operating room.**

**Rating:** 6
**Confidence:** 4

**Review:**

1.It is recommended that authors summarize the contribution of the method proposed in this article in the introduction of the article to highlight the research advantages of this article.
2.In visual-based screen segmentation and text recognition, what strategies should be implemented to enhance accuracy, robustness, and efficiency when dealing with complex screen layouts, variability, and frequent changes in user interfaces and documents?
3.Figures 1, 3, 8, 9, and 12 are unclear, it is recommended to redraw them.
4.The formatting of the references needs to be carefully reviewed and adjusted.

---

### Decision · Program_Chairs · 2024-09-06

Accept (Oral)